# Structural Features and Photoelectric Properties of Si-Doped GaAs under Gamma Irradiation

**DOI:** 10.3390/nano10020340

**Published:** 2020-02-17

**Authors:** Ye Shen, Xuan Fang, Xiang Ding, Haiyan Xiao, Xia Xiang, Guixia Yang, Ming Jiang, Xiaotao Zu, Liang Qiao

**Affiliations:** 1School of Physics, University of Electronic Science and Technology of China, Chengdu 610054, China; shenye_uestc@sina.com (Y.S.); mjianglw@gmail.com (M.J.); xtzu@uestc.edu.cn (X.Z.);; 2State Key Laboratory of High Power Semiconductor Lasers, School of Science, Changchun University of Science and Technology, Changchun 130022, China; fangxuan110@hotmail.com; 3Institute of Nuclear Physics and Chemistry, China Academy of Engineering Physics, Mianyang 621900, China; biansechong@163.com

**Keywords:** GaAs, gamma radiation, structural features, photoelectric property

## Abstract

GaAs has been demonstrated to be a promising material for manufacturing semiconductor light-emitting devices and integrated circuits. It has been widely used in the field of aerospace, due to its high electron mobility and wide band gap. In this study, the structural and photoelectric characteristics of Si-doped GaAs under different gamma irradiation doses (0, 0.1, 1 and 10 KGy) are investigated. Surface morphology studies show roughen of the surface with irradiation. Appearance of transverse-optical (TO) phonon mode and blueshift of TO peak reflect the presence of internal strain with irradiation. The average strain has been measured to be 0.009 by Raman spectroscopy, indicating that the irradiated zone still has a good crystallinity even at a dose of 10 KGy. Photoluminescence intensity is increased by about 60% under 10 KGy gamma irradiation due to the strain suppression of nonradiative recombination centers. Furthermore, the current of Si-doped GaAs is reduced at 3V bias with the increasing gamma dose. This study demonstrates that the Si-doped GaAs has good radiation resistance under gamma irradiation, and appropriate level of gamma irradiation can be used to enhance the luminescence property of Si-doped GaAs.

## 1. Introduction

Gallium Arsenium (GaAs), as a direct band gap semiconductor material, shows high application value in optoelectronic devices and nuclear microwave devices due to its suitable band gap, high electron/hole mobility, and photoelectric conversion efficiency [1,2,3,4,5]. The solar cells, photodetectors, light-emitting diodes and field-effect transistors based on GaAs have been widely applied in aerospace industry. However, in the outer space environment, the cosmic rays such as high-energy protons, high-energy electrons, X-rays, neutrons and gamma rays have significant impacts on the performance of electronic devices and instruments [6,7,8,9,10,11]. In order to improve the survivability and reliability of devices under various radiation environments, it is necessary to study the photoelectric behavior of electronic devices based on GaAs under irradiation conditions.

Over the past few decades, much efforts were devoted to investigating the structural features and photoelectric behavior of GaAs under different radiation environments. Peercy et al. reported that the lattice strain can be detected by comparing the Raman scattering phonon mode of GaAs before and after Xe ion irradiation [12]. In terms of electrical properties, Jayavel et al. studied the shallow level defect concentrations of the non-irradiated and N^+^ ion-irradiated Si doped GaAs diodes, and the concentrations were found to decrease from 1.4 × 10^17^ to 4.29 × 10^16^ cm^−3^ due to trapping of charge carriers by the irradiation induced defects [13]. Lai et al. found that there were nearly 40% drop in free-carrier concentration of GaAs under the effect of ^60^Co γ-irradiation with dose of 300 Mrad. This is due to additional acceptor traps created after irradiation, which act as recombination centers to compensate the shallow donors in the undoped n-GaAs devices [14]. Similar phenomenon was observed by Stievenard et al., who reported that the free-carrier concentration of GaAs decreases as the electron fluence increases [15]. As for optical properties, Carin et al. investigated the luminescence of sikicon-doped (n-doped) and zinc-doped (p-doped) GaAs irradiated by 0.3 GeV calcium and 0.7 GeV zinc ions various fluence, and they found the photoluminescence signal decreases with increasing fluence due to the production of nonradiative defects [16]. Khanna et al. found that the photoluminescence (PL) intensity of GaAs increases under the irradiation of 10^10^ e/cm^2^ electrons and 3 × 10^13^ n/cm^2^ neutrons, which is due to the fact that low flux irradiation reduces the hole trap concentration [17].These studies demonstrate that radiation environments have significant impacts on the photoelectric properties of GaAs. Theoretically, Nordlund et al. studied defect production in GaAs under low-energy self-recoils and 6 MeV He ion irradiation by using molecular dynamics simulations. Their results showed that the majority of created defects are clusters, and the isolated defects are mainly interstitials [18]. Mazouz et al. used numerical simulation to model the performance degradation of a single-junction GaAs solar cell under 1 MeV electron irradiation, and they found that the GaAs solar cell is sensitive to 10^16^ cm^−2^ electron irradiation fluence. The electron defects E3, E4 and the hole defect H4 created by electron irradiation are the most responsible for the degradation of short-circuit current [19].

Investigation of the optoelectronic properties of GaAs under different radiation environments allows us to understand the irradiation thresholds of GaAs-based electronic devices operating in extreme environments, which can ensure their stability and reliability under extreme conditions and is especially important in the aerospace industry. In the literature, although the radiation effects of GaAs have been studied extensively, these studies are mainly focused on ion and electron irradiation and the influence of γ-irradiation on Si-doped GaAs has rarely been reported. Gamma rays are generally considered to be produced by fusion of stellar cores and widely exist in the space environment. It is thus necessary to investigate how the GaAs optoelectronic devices respond to gamma irradiation. In this study, the structural and photoelectric characteristic of Si-doped GaAs irradiated by γ-irradiation at different doses (0, 0.1, 1 and 10 KGy) are studied. It is interesting to find that the Si-doped GaAs is resistant to high-dose gamma radiation, and its luminescence property can be improved by the radiation.

## 2. Experimental Methods

The GaAs sample is grown by P600 molecular beam epitaxy (MBE) system (DCA Instruments, Turku, Finland). A GaAs buffer layer is first grown on a (001) oriented semi-insulating GaAs substrate (10 × 10 cm^2^) by homogeneous epitaxy, and then an ohmic contact layer of 1 μm thickness n-type GaAs is grown on the buffer layer, where the Si doping concentration is 2 × 10^18^ cm^−3^. The structure diagram of Si-doped GaAs is shown in Figure 1. The growth temperature is about 670 °C, and the As/Ga flux ratio is 20. The γ-irradiation of all samples is performed for the entire surface of the films, in a ^60^Co irradiator, with different doses of 0, 0.1, 1 and 10 KGy and the average dose rate of 0.0846 Gy/s. The Being CSPM 4000 Atomic Force Microscope (AFM) (Being Nano-Instruments, Guangzhou, China) is employed to characterize the morphology and surface roughness of Si-doped GaAs before and after irradiation. Raman spectroscopy and Room temperature Photoluminescence (PL) spectra are also measured in a WITec alpha300 R working with a green laser at 488 nm (WITec Wissenschaftliche Instrumente und Technologie GmbH, Ulm, Germany). The I-V measurements are carried out by using a Keithley 4200 measurement unit (Keithley Instruments & Products, Mansfield, TX, USA).

## 3. Results and Discussion

### 3.1. The Morphological Structures of Si-Doped GaAs under Gamma Radiation

AFM is used to characterize the morphological structures of Si-doped GaAs under different gamma radiation doses. Figure 2a shows the three-dimensional AFM images of non-irradiated Si-doped GaAs, which are scanned in the range of 3 × 3 μm^2^. The mean roughness (RA) is 0.199 nm and the root means square (RMS) roughness is 0.255 nm. The roughness of the non-irradiated GaAs is on the order of 10^−1^ nm, indicating that the surface is very smooth. The surface morphology of Si-doped GaAs at 0.1, 1 and 10 KGy are also scanned in the range of 3 × 3 μm^2^, and their corresponding three-dimensional AFM images are presented in Figure 2b–d. The RMS of Si-doped GaAs under different gamma doses are summarized in Table 1. At the dose of 0.1 KGy, the RMS is 0.336 nm, which is comparable to the value of 0.225 nm for un-irradiated GaAs, indicating that the surface is still compact and flat. The surface roughness increases to 3.16 nm under the dose of 1 KGy, which shows clearly distinct features of craters and hills. When the irradiation dose reaches 10 KGy, the surface roughness is increased to 3.73 nm, where the craters are merged due to their overlapping proximity, indicating that the disorder of the Si-doped GaAs surface layer increases further. The experimental observation that the surface roughness increases with increasing γ radiation dose suggests that the surface microstructure of GaAs has been modified by γ radiation, and the irradiated sample may exhibit different photoelectric properties from the pristine phase.

### 3.2. Lattice Disorder and Crystallinity of Si-Doped GaAs under Gamma Radiation

The Raman spectra of Si-doped GaAs at gamma doses of 0, 0.1, 1 and 10 KGy are shown in Figure 3. For unirradiated GaAs, the longitudinal-optical (LO) phonon of GaAs is localized near 289.7 cm^−1^ and transverse-optical (TO) phonon is located at 269 cm^−1^. In principle, according to selection rules analyzed by symmetry-related Raman dispersion tensor, for zinc-blende structured GaAs with (001) crystallographic orientation, the LO phonon is allowed to appear in the Raman spectrum of backscattering geometry, while the TO phonon is generally symmetry-forbidden [20,21]. However, in reality, many factors such as strain or defect during doping or epitaxy growth might introduce momentum non-conservation, which can relax the Raman scattering selection rule, leading to the appearance of prohibited TO-GaAs mode [22,23]. Therefore, for most experimental GaAs epitaxial films, both weak TO and strong LO peaks can exist. We also observe LO and TO phonon frequency at 290.7 cm^−1^ and 269 cm^−1^, respectively, for intrinsic non-doped GaAs epitaxial films with I_TO/LO_ ratio of 0.907, which is consistent with previous reports [21]. In contrast, the unirradiated Si-doped GaAs film shows much stronger TO phonon peak than LO with an I_TO/LO_ ratio of 1.372, suggesting the presence of surface defects which help to break the Raman scattering selection rule.

Upon γ-irradiation, it is clearly seen that TO phonon peak not only increases in intensity but also demonstrates a blue shift at larger irradiation dose. Table 2 summarizes the transverse and longitudinal optical phonon frequency of Si-doped GaAs under 0, 0.1, 1 and 10 KGy γ-radiation, as well as the TO/LO peak intensity ratio. It is seen that as the γ-irradiation dose increases, the TO peak intensity and TO/LO peak intensity ratio increases, indicating presence of more surface defects and degree of lattice disorder that responsible for stronger level of Raman selection rule breaking, leading to the increase of TO peak intensity. From irradiation physics point of view, gamma irradiation is in distinctive contrast to ion irradiation (such as electron, neutron, ion, etc.), while ion irradiation introduces both ionization and displacement defects, gamma irradiation only introduces ionization defect as direct momentum contribution from photons are considered to be very weak. Therefore, the ionization defects are not strong enough to significantly destroy the long-range crystalline order. As a result, the film structure is not significantly affected as evidenced by the AFM results in Figure 2. However, these ionization defects can introduce a small level of internal strain, which is further evidenced by the fact that both LO and TO phonon peak show a clear trend of blueshift, as shown in Figure 3 and Table 2. It is noted that the blueshift is very slight. For example, at the dose of 10 KGy, the position of the LO and TO phonons of Si-doped GaAs are shifted downward by only 2.8 cm and 3.8 cm, respectively. Similar epitaxial strain induced TO shift is also observed by Signorello et al. [24] in GaAs nanowire system.

Raman spectroscopy can also be used to estimate the internal strains of epitaxial film in a nondestructive way [25]. By measuring the frequency shift of LO or TO phonon, the average stress (σ) can be obtained from the following equation:(1)$$\sigma =\frac{2\omega_{TO}\Delta \omega_{TO}}{\left(p+q\right)S_{11}+\left(p+3q\right)S_{12}}$$
where $\omega_{\mathrm{TO}}$ is TO phonon frequency, ${\Delta \omega }_{TO}$ is the TO phonon frequency shift, S_*ij*_ is the elastic compliance coefficient [26,27], *p* and *q* are the deformation potential energy, respectively. Under stress, the in-plane strain can be calculated by ε*_p_* = (S_11_ + S_12_)*σ*. By using the parameters in Table 3, the relationship between TO phonon frequency shift and strain as well as stress is obtained to be Δ*ω_TO_* = −2.45*σ* = −302ε*_p_*. The $\Delta \omega_{TO}$ of irradiated GaAs relative to the non-irradiated sample and the obtained strain are summarized in Table 4. For the irradiated GaAs, the stresses *σ* are all larger than zero due to blueshift of $\omega_{TO}$, indicating the presence of strain that becomes larger with increase of radiation dose. Figure 4 shows the variation of lattice strain ($\varepsilon_{p}$) of GaAs with the gamma dose. It can be seen that $\varepsilon_{p}$ increases with the increasing dose and the maximum strain of 0.009 is obtained for irradiation dose of 10 KGy. As compared with the maximum strain of 0.038 for unrelaxed bulk GaAs, the small strain of irradiated GaAs obtained in this work indicates that the crystallinity of the irradiated zone is still good [25]. Therefore, although lattice strain and disorder of the surface layer gradually increases with increasing gamma dose, significant structural deterioration still does not occur even under 10 KGy gamma irradiation. These results further demonstrate that Si-doped GaAs shows good radiation resistance under gamma irradiation.

### 3.3. Photoelectric Characteristics of Si-Doped GaAs under Gamma Radiation

To investigate the impact of γ-radiation on the electronic properties, PL spectroscopy of the Si-doped GaAs under 0.1, 1 and 10 KGy at room temperature (300 K) are shown in Figure 5. The transition peak of 1.459 eV is considered as resulting from band-to-band transition with a high doping concentration of 2 × 10^18^ Si/cm^3^ at room temperature, which is consistent with the literature [28]. It can be observed that the intensity of the light-emitting peak is enhanced under low dose γ irradiation. When the irradiation dose reaches 1 KGy, the luminous intensity is increased by 37%, as compared with the non-irradiated sample, and the value is increased by about 60% at the dose of 10 KGy. This phenomenon may be understood from aforementioned surface ionization defect point of view. As discussed, the unirradiated Si-doped GaAs epitaxial film contains certain amount of surface defects, which might act as nonradiative recombination centers during photoluminescence excitation. The presence of low dose gamma irradiation is shown to introduce internal strain as evidenced from Raman spectroscopy. This low-dose irradiation-induced strain is expected to not only relieve the existing nonradiative recombination centers but also create extra new recombination centers at the surface region, thus leading to enhancement of PL response [29]. Similar low dose γ-radiation induced enhancement of PL has been reported in single crystal n-type GaAs [30]. Meanwhile, Khanna et al. also found that both electron and neutron irradiation would reduce a hole trap concentration at low fluences, and the decrease of the trap concentration is accompanied by an increase of PL intensity at lower fluences (10^10^ electron/cm^2^ or 3 × 10^13^ neutron/cm^2^) [17]. We reiterate that the dose level is very important; the maximum dose in this study, 10KGy (~10^6^ rad), is a moderate value, whereas a very high dose level, e.g., >10^7^ rad, is shown to significantly decrease the PL response [30] due to severe deep-level defect formation. Therefore, from the materials’ perspective, films have more irregularities of atomic structure and have weaker bonds at the surfaces, phase boundaries or interfaces; thus, some perturbation such as internal strains due to external irradiation could induce configuration modification, leading to a lower energy state. In this study, the original defects in pristine unirradiated Si-doped GaAs film may exist in the form of non-radiative recombination centers, and subsequent gamma irradiation is beneficial to remove these non-radiative recombination centers, leading to significant enhancement of luminescence intensity. Furthermore, from Figure 5, we also find that the PL peak position show litter dependence with γ-irradiation dose. For example, the light emission peak of GaAs at the dose of 10 KGy increases slightly, nearly 3 meV shift from 1.455 eV to 1.458 eV, with respect to the non-irradiation sample. Very weak dependence of GaAs band gap on the epitaxial strain has also been reported experimentally by Signorello et al. [24]. This fact suggests that the γ-irradiation only modifies the surface defects (nonradiative recombination centers) but does not affect the bulk electronic transition; thus, the fundamental band gap remains almost the same. Moreover, weak shift of PL peak is also consistent with the strain remaining below 1% determined by Raman spectroscopy.

Finally, I-V curve of GaAs is presented in Figure 6. Under 3 V bias voltage, the current of the sample decreases with the radiation doses increasing from 0 to 10 KGy. It is seen that for unirradiated GaAs epitaxial film, it shows a rectified, non-ohmic behavior, possibly due to the configuration of the top electrode (in this work, the top electrode is wire-bonded through Al wires). It has been reported that a good coherent metal electrode pad is helpful to demonstrate the intrinsic Ohmic behavior [31,32]. Upon γ-radiation, it is seen clearly that the electrical behavior still keeps the non-Ohmic behavior but degrades with the increase of γ-ray dose. For the maximum dose of 10 kGy, the current decreased about two orders of magnitude compared to the unirradiated sample (see logarithm scale plot in inset of Figure 6). This level of property degradation is smaller than early report by Deshmukh et al. that ion irradiation (10^13^ ~ 10^14^ ions/cm^2^) of GaAs device show typical 3 ~ 4 order of magnitude degradation in electric current [33]. The deterioration of the electrical properties is mainly ascribed to the irradiation induced ionization defects that significantly suppress quantity and mobility for the carriers generated by Si-doping, leading to drop of the overall electrical current [34].

## 4. Conclusions

In this study, we have investigated the structural and photoelectric characteristics of Si-doped GaAs under different gamma radiation doses (0, 0.1, 1 and 10 KGy). The results show that the roughness of Si-doped GaAs film increases with the increasing irradiation dose, while Raman scattering analysis suggests that the sample still has a good crystallinity. Besides, the luminescence intensity of Si-doped GaAs film increases by about 60% under 10 KGy gamma radiation, indicating that gamma radiation may help to remove the non-radiation composite center of the GaAs layer. However, the current of Si-doped GaAs is significantly reduced due to the decrease of carrier concentration and carrier mobility. In conclusion, the Si-doped GaAs is resistant to gamma irradiation even at a high dose of 10 KGy, and gamma-ray treatment is suggested to be an effective way to enhance the luminescent properties of Si-doped GaAs.

## Figures and Tables

**Figure 1 nanomaterials-10-00340-f001:**
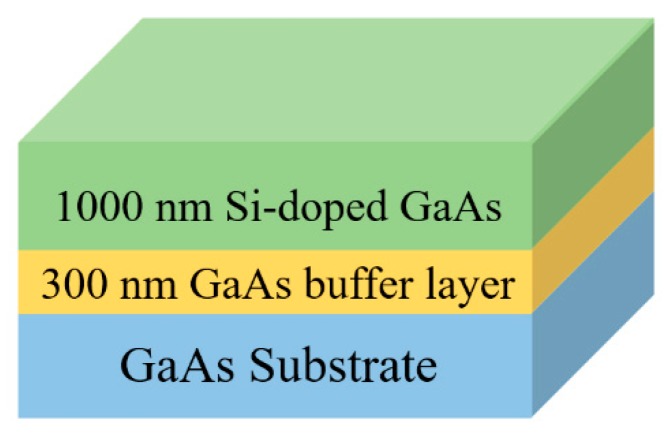
The schematic diagram of Si-doped Gallium Arsenium (GaAs) sample.

**Figure 2 nanomaterials-10-00340-f002:**
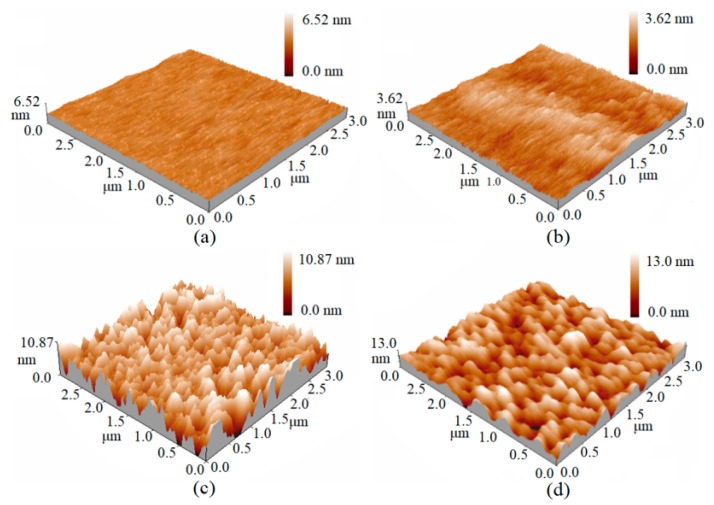
The 3D Atomic Force Microscope (AFM) images of Si-doped GaAs before (**a**) and after (**b**–**d**) γ-radiation. The gamma doses are 0, 0.1, 1 and 10 KGy.

**Figure 3 nanomaterials-10-00340-f003:**
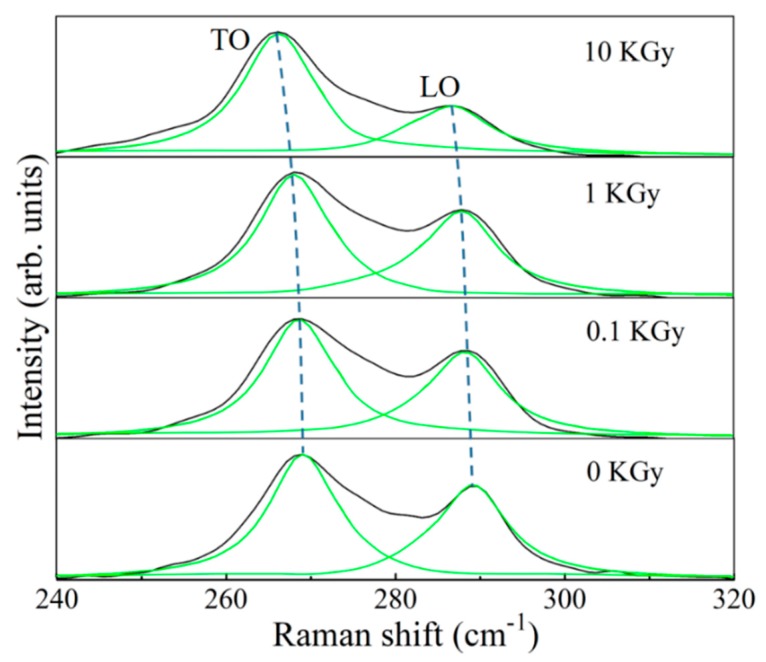
The Raman spectra of the Si-doped GaAs under different γ-radiation doses (0, 0.1, 1 and 10 KGy) along with the spectrum of the undoped GaAs substrate (black line) as a reference.

**Figure 4 nanomaterials-10-00340-f004:**
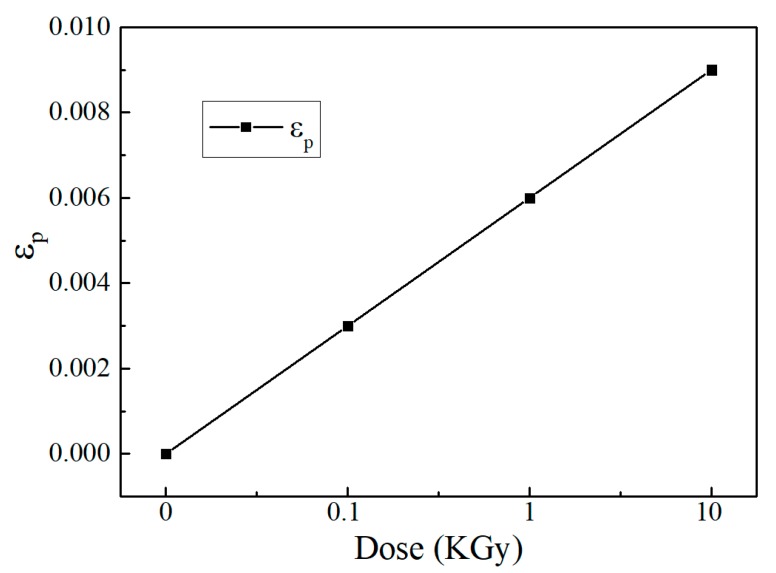
The variation of lattice strain ($\varepsilon_{p}$) of GaAs as function of gamma dose.

**Figure 5 nanomaterials-10-00340-f005:**
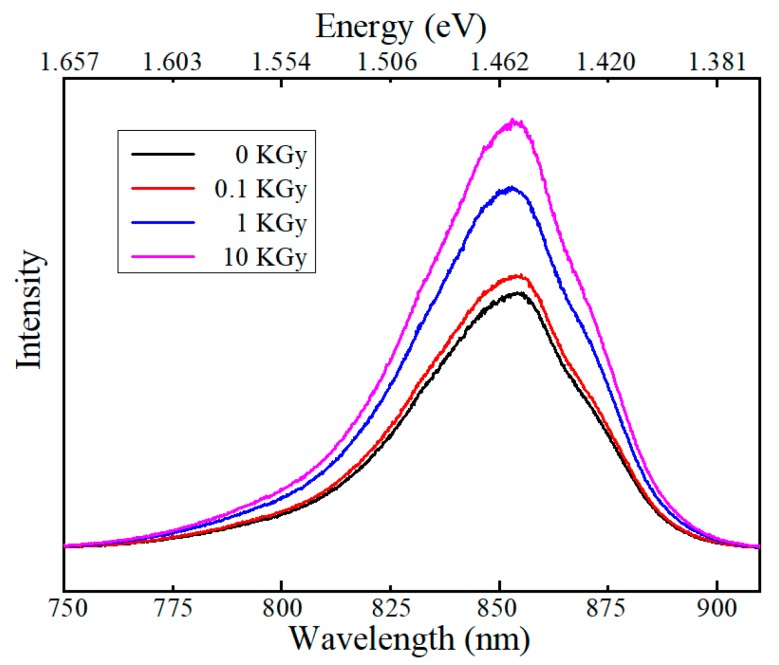
The photoluminescence spectrum of GaAs at room temperature under different gamma irradiation doses (0, 0.1, 1 and 10 KGy).

**Figure 6 nanomaterials-10-00340-f006:**
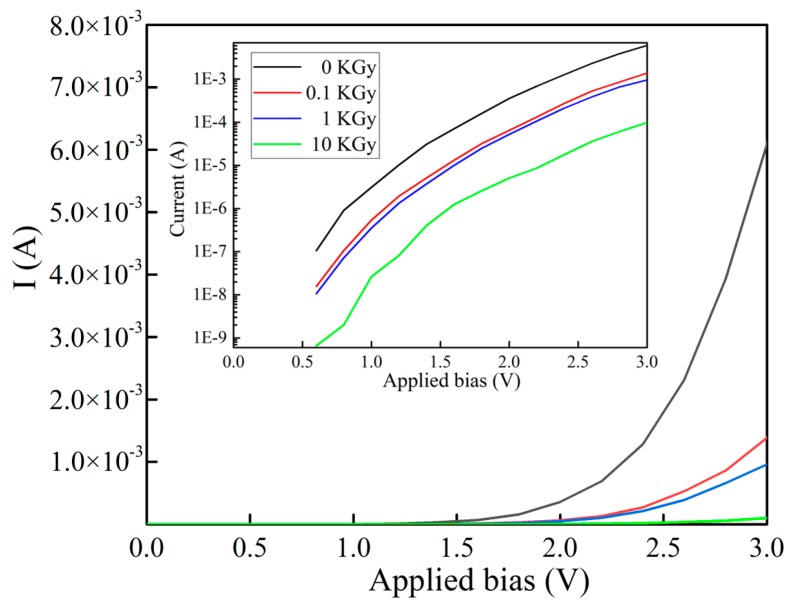
The I-V curve of Si-doped GaAs at 3 V bias voltage under different γ-radiation doses (0, 0.1, 1 and 10 KGy). Inset shows the same graph in logarithm scale.

**Table 1 nanomaterials-10-00340-t001:** The root means square (RMS) roughness of Si-doped GaAs under 0, 0.1, 1 and 10 KGy γ-radiation.

	Dose (KGy)			
RMS (nm)	0	0.1	1	10
	0.332	0.336	3.16	3.73

**Table 2 nanomaterials-10-00340-t002:** The transverse-optical phonon frequency (ω_GaAs TO_) and longitudinal-optical phonon frequency (ω_GaAs LO_) of Si-doped GaAs under 0, 0.1, 1 and 10 KGy γ-radiation. The I_TO/LO_ is the phonon intensity ratio of transverse-optical phonon and longitudinal-optical phonon.

Dose (KGy)	ω_GaAs TO_ (cm^−1^)	ω_GaAs LO_ (cm^−1^)	I_TO/LO_
0	269.008	289.714	1.372
0.1	268.065	288.774	1.425
1	267.122	288.774	1.475
10	266.179	285.954	2.754

**Table 3 nanomaterials-10-00340-t003:** The elastic parameters S_11_ and S_12_ of Si-doped GaAs and deformation potentials *p*, *q* for TO phonons used in the study. The parameters are cited from Reference [14].

Lattice Parameter (nm)	S_11_(10^−2^ Gpa^−1^)	S_12_(10^−2^ Gpa^−1^)	[(*p* − *q*)/ω_0_^2^]_TO_	[−(*p* + 2*q*)/6ω_0_^2^]_TO_
0.5653	1.175	−0.365	0.3	1.11

**Table 4 nanomaterials-10-00340-t004:** The phonon frequency shift ($\Delta \omega_{TO}$) and the strain (ε*_p_*) for Si-doped GaAs under 0, 0.1, 1 and 10 KGy γ-radiation.

Dose (KGy)	Δω_GaAs *TO*_ (cm^−1^)	ε*_p_*
0	0	0
0.1	−0.943	0.003
1	−1.886	0.006
10	−2.829	0.009

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
