# Peer review of "Structural Features and Photoelectric Properties of Si-Doped GaAs under Gamma Irradiation"

_nanomaterials, 2020, doi:10.3390/nano10020340_

Round 1
Reviewer 1 Report
This paper is a study on the causes of physical property changes in GaAs-based semiconductor devices exposed to Gamma-ray exposure in space-like environments. Although related papers have been published, some additional experimental proposals have been proposed using Raman. Therefore, if the following questions are revised, it is possible to publish them.
1) About the crystal structure change, maybe the authors can prove the change of lattice constant based on the X-ray diffraction or electron diffraction.
2) About the PL measurement, How about the main peak of GaAs near band edge.? The author can supply the data.
3) Comments about the possible protection layer to block the degradation of GaAs by Gamma ray.
Reviewer 2 Report
Abstract:
Say "GaAs has been demonstrated..." not "The Gaas has been ...". Revise the structure (Change ", which has been" -> ". It has been")
Revise English throughout.
1 Intro:
The review could be more focussed. The Presentation summarises elements of the literatue without spending enough time linking these element to the work being intoduced.
The last pargraph goes some way towards addressing this.
2 experimental
No comment.
3 Results
Figure 1 : 2D / 3D does not add infomation. Consider using only the 3D for comparison with figure 2.
Required: define "injection region". The all important stress parameter is defined from substrate lattice parameter and "injection layer" lattive parameter which presumbly means the irradiated material, and one guesses this refers to injection of radiation. However since "injection layer" has specific meaning in semiconductor device physics, it should be clarified (and ideally a better terminology used but this is optional).
Required : a figure showing the structure : substrate, and irradiated volume, and contact layers (plural : only one contact is mentioned in the text).
Required : State the physical dimensions of the sample in the text, referring to the required figure of substrate and irradiated volume.
This figure and these details on structure might be best placed in "experimental" which at present is succinct.
Required : The PL increase with dose is only discussed phenomenologically and with insufficient precision. The reason this is striking is that the discussion of ordering iscompletely separate from the discussion of incersing disorder just above int eh Raman discussion. Since at present this is a contradiction, it must be resolved before the paper can be published.
To address this :
Explain precisely how radiation damage can improve crystallinity, citing details including as the mechanisms of re-ordering such as modifications to crystalline point defects.
Relate this to the density of defects before irradiation in you GaAs substrate : claims of improved crystallinity must start from an understanding of the state before irradition.
Most importantly : explain the apparent contradiction with the radiation induced strain observed in the Raman measurements.
Comment : The paper would benifit from stating the effect of strain on the bandgap. The absence of a shift in PL peak suggests that this is low, which is roughly consistent with the strain remaining below 1% - nevertheless, it poses a question in the reader's mind "have they considered strain effects on band structure". Stating the expected effects of the stated strain on teh GaAs bandgap would impove the paper.
Comment : The dark current measurements as presented pose questions:
1) "U/V" is undefined as axis label : presumably "applied bias" is meant.
2) As presented, the data shows non-ohmic behaviour. Given the linear scale, and the presentation in current rather than current density it is unfortunately impossible to make much of this data other than "forwards bias current is decreased with increasing irradiation"
Presentation in current density and a log/long scale, with correct axis labels, would improve this.
Required change : The non-ohmic current-voltage curve must be explained in the light of the claimed ohmic contacts.
This includes some discussion of the "ohmic contacts" claimed earlier in the paper.
The question is where the apparent non-ohmic behaviour comes from, and most importantly, is it related to the irradiation.
There is detailed discussion at the end of the results and discussion section concerning, to cut a long story short, a modification of effective doping due to structural damage : what comes to mind therefore is the formation of either an insulating layer and / or the fomation of a semiconductor doping junction which could explain the non-ohmic IV behaviour and apparent transition to an insulater at higher dose.
Requirement : the discussion must be developed since the existing discussion falls short.
4 Conclusions :
~~~~~~~~~~
See below.
Overall comments :
This interesting work needs a rewrite. The basic claim of the paper is not demonstrated convincingly.
More seriously, the results, as presented, lead to some contradiction.
First the morphology study clearly demonstrated material damage as expected.
This contradicts subsequent claims that irradiation is in effect annealing the material.
Secondly the claim of increased material quality contradicts the observed increase in strain (even if this remains below 1%).
Furthermore, given the questions referred to above concerning non-ohmic IV, possible pn juntions and maybe space-charge layers in the device, the conclusion that radiation damage improves material quality is not demonstrated.
The paper must be re-written to remove the apparent contradictions as follows :
Define the strucrure, demonstrate the ohmicity of the contacts Give details of claimed improvement in crystallinity. Relate to density of defects i GaAs before irradiation and after. Explain the apparent contradiction with the increasing strain as a function of irradiation - Explain the non-ohmic dark current curves (possible explanations in doping modification in the irradiated zone). Further required changes : see all the notes on the text.
This can be an interesting piece of work once these contrdictions and questions have been addressed.
Round 2
Reviewer 1 Report
The author prepared reasonable revised version by considering comments raised by reviewers. So, the manuscript can be published in Nanomaterials.
Reviewer 2 Report
The revision is satisfatory. While some additional improvements might be suggested, since some responses to the first review are "sufficient" rather than "excellent", the overall paper is nevertheless much improved.
This can be published as is.